# Targeted Co-Delivery of Gefitinib and Rapamycin by Aptamer-Modified Nanoparticles Overcomes EGFR-TKI Resistance in NSCLC via Promoting Autophagy

**DOI:** 10.3390/ijms23148025

**Published:** 2022-07-21

**Authors:** Yuhong Liu, Xiaoyong Dai, Shengwei Jiang, Mulan Qahar, Chunyan Feng, Dongdong Guo, Lijun Wang, Shaohua Ma, Laiqiang Huang

**Affiliations:** 1Shenzhen Key Laboratory of Gene and Antibody Therapy, Center for Biotechnology and Biomedicine, State Key Laboratory of Chemical Oncogenomics, State Key Laboratory of Health Sciences and Technology, Precision Medicine and Healthcare Research Center, Tsinghua-Berkeley Shenzhen Institute (TBSI), Institute of Biopharmaceutical and Health Engineering, Shenzhen International Graduate School, Tsinghua University, Shenzhen 518055, China; liuyh18@tsinghua.org.cn (Y.L.); dai-xy18@mails.tsinghua.edu.cn (X.D.); shengweijiang2016@163.com (S.J.); mulanqahar@126.com (M.Q.); 14211030052@fudan.edu.cn (C.F.); dongdong-guo@foxmail.com (D.G.); wang.lijun@sz.tsinghua.edu.cn (L.W.); ma.shaohua@sz.tsinghua.edu.cn (S.M.); 2Department of Chemistry, Tsinghua University, Beijing 100084, China

**Keywords:** gefitinib, cell autophagy, rapamycin, drug combination therapy, targeted drug delivery

## Abstract

Acquired drug resistance decreases the efficacy of gefitinib after approximately 1 year of treatment in non-small-cell lung cancer (NSCLC). Autophagy is a process that could lead to cell death when it is prolonged. Thus, we investigated a drug combination therapy of gefitinib with rapamycin—a cell autophagy activator—in gefitinib-resistant NSCLC cell line H1975 to improve the therapeutic efficacy of gefitinib in advanced NSCLC cells through acute cell autophagy induction. Cell viability and tumor formation assays indicated that rapamycin is strongly synergistic with gefitinib inhibition, both in vitro and in vivo. Mechanistic studies demonstrated that EGFR expression and cell autophagy decreased under gefitinib treatment and were restored after the drug combination therapy, indicating a potential cell autophagy–EGFR positive feedback regulation. To further optimize the delivery efficiency of the combinational agents, we constructed an anti-EGFR aptamer-functionalized nanoparticle (NP-Apt) carrier system. The microscopic observation and cell proliferation assays suggested that NP-Apt achieved remarkably targeted delivery and cytotoxicity in the cancer cells. Taken together, our results suggest that combining rapamycin and gefitinib can be an efficacious therapy to overcome gefitinib resistance in NSCLC, and targeted delivery of the drugs using the aptamer-nanoparticle carrier system further enhances the therapeutic efficacy of gefitinib.

## 1. Introduction

Non-small-cell lung cancer (NSCLC) is a common subtype of lung cancer, accounting for almost 85% of all cases. Growing evidence has revealed that epidermal growth factor receptor (EGFR) is highly expressed in NSCLC cells, and its activation via ligand binding (epithermal growth factor) can induce phosphorylation via tyrosine kinase activity and consequently promote tumor cell proliferation [1]. Gefitinib (ZD1839 or Iressa) is the most remarkable first-generation EGFR tyrosine kinase inhibitor (TKI), applied as a first-line NSCLC therapy. Mutations located in the *EGFR* tyrosine kinase region (exon 19 deletion or exon 21 *L858R* mutation) have been closely linked to patients’ positive response to gefitinib [2,3]. Despite the initial positive response, most patients inevitably develop acquired resistance, accompanied by severe side effects due to the use of high-dose gefitinib. In over 60% of patients, the mechanisms of acquired resistance involve a secondary *EGFR* mutation characterized by a threonine to methionine substitution at position 790 (*T790M* mutation) [3]. Meanwhile, multiple studies have revealed additional mechanisms of gefitinib resistance, including the activation of EGFR downstream or parallel mechanisms, epithelial–mesenchymal transition, and cell autophagy [4,5,6,7,8,9]. Although second- and third-generation EGFR-TKIs have been developed to overcome the aforementioned problems, acquired resistance emerges eventually [10]. Therefore, alternative strategies should be considered.

Drug combination therapy has long been accepted as a potent anti-cancer management to overcome tumor resistance. Although accumulating evidence has highlighted that EGFR-TKI treatment can induce persistent NSCLC cell autophagy, its role in subsequent cell survival is still considered to be contextual [11,12]. Since activation of the autophagic pathway beyond a certain threshold may directly promote cell death [13], we infer that a cell autophagy activator that can augment gefitinib-induced cell autophagy may be a desirable candidate for combination therapy against gefitinib-resistant NSCLC. Rapamycin is a well-studied cell autophagy activator that functions as an inhibitor of the mammalian target of rapamycin (mTOR), which is a negative regulator of cell autophagy located downstream of the EGFR signaling pathway. Furthermore, since mTOR serves as a regulator of cell cycle and growth in response to altered nutrient levels in multiple types of cancer cells, rapamycin is effective against almost all types of solid tumors [14,15]. Therefore, we speculate that rapamycin can not only improve the therapeutic efficacy of gefitinib through the induction of autophagic cell death but also kill NSCLC cells through its anti-tumor activity.

Regarding drug combination therapy, drug toxicity is a significant problem. Moreover, if drugs with dissimilar pharmacokinetic profiles are administrated in a free molecular form, the combinational agents cannot be effectively delivered to the target sites with desirable distribution and duration [16]. In this context, studies on co-delivery and targeted delivery using nanomedicine have opened extraordinary opportunities for facilitating drug retention and uptake by tumor cells. For instance, ligand-conjugated chitosan nanoparticles have been widely applied in cancer diagnosis and therapy owing to the outstanding cell membrane affinity and biocompatibility of chitosan and its remarkable potential for targeted drug delivery [17,18]. Recently, oligonucleotide aptamers have garnered much popularity in targeted drug delivery, since they can bind the target proteins with high affinity and low immunogenicity [19]. In a previous study, the high affinity of an anti-EGFR aptamer toward EGFR and its successful application offered novel insights into the development of an anti-EGFR aptamer-functionalized chitosan nanoparticle (NP-Apt) carrier system [20].

To this end, in the present study, we experimentally demonstrated a strategy to overcome gefitinib resistance in NSCLC through nanomedicine-based drug combination therapy. The combination of gefitinib and rapamycin efficiently induced autophagic cell death. Additionally, we examined the therapeutic efficacy of this combination therapy in the human NSCLC cell line H1975 (harboring *L858R* and *T790M* mutations) and investigated whether it links with autophagy-mediated cell death. To minimize side effects, we further constructed an NP-Apt carrier system and subsequently evaluated its cell toxicity and targeted delivery efficiency in vitro. Overall, the proposed drug combination therapy appears efficient in the treatment of gefitinib-resistant NSCLC, and the developed aptamer-modified nanoparticles show great potential in improving the targeted delivery efficiency and therapeutic efficacy of the combined medication.

## 2. Results

### 2.1. Combination Therapy of Rapamycin and Gefitinib Attenuates H1975 Cell Viability and Xenograft Tumor Growth Significantly

To evaluate the therapeutic efficacy of the combination therapy, we treated 2D- and 3D-cultured H1975 cells with different concentrations of gefitinib, rapamycin, or a combination of gefitinib and rapamycin, and then analyzed their cell viability. The 3D cell culture model used in this study provides cells with an environment that induces them to behave in a natural condition, as shown in our previous studies [21,22]. Here, they displayed as spheres with a diameter of 250 ± 50 μm (Appendix A). The half-maximal inhibitory concentration (IC_50_) value of gefitinib in 3D cultures was 10.64 μM, which is higher than that in 2D cultures (6.95 μM). A reasonable explanation for this difference is that the penetration of drugs in a 3D culture system often undergoes interference by an abundant extracellular matrix, and thus, such penetration becomes more difficult. Next, we evaluated cell viability under drug combination treatment by keeping gefitinib at 5 μm, since 5 μm is slightly lower than its IC_50_ in H1975 cells, and it could help us identify the efficacy of the combinational therapy. As shown in Figure 1A, it was found that the combination treatment of gefitinib and rapamycin significantly decreased the viability of H1975 cells in the 2D-cultured system compared to treatment with only gefitinib or rapamycin. Similar results were observed in 3D cultures (Figure 1B and Appendix A). The combinational index in 2D and 3D cultures was between 0.9 and 1.1 and below 0.9, indicating that the gefitinib and rapamycin worked additively in 2D cultures, and synergically in 3D cultures. Further, we evaluated the synergic score through SynergyFinder 2.0 and the synergy score was over 10 (Appendix A), indicating the interaction between gefitinib and rapamycin in 2D/3D cultures is synergistic [23]. Next, we treated H1975 cells with gefitinib (5 μM), rapamycin (10 μM), and their combination to identify their colony formations. The concentration was determined because gefitinib (5 μM) + rapamycin (10 μM) has significant efficiency for both 2D and 3D cultures. Figure 1C suggested that the combination treatment of gefitinib and rapamycin exhibited significantly stronger effects on colony formation inhibition than in the control or single-drug group. Subsequently, we further examined the therapeutic efficacy of the combination treatment of gefitinib and rapamycin in gefitinib-resistant tumors. Our results demonstrated that rapamycin potentiates the therapeutic efficacy of gefitinib in the H1975 xenograft model (Figure 1D). H&E staining displayed the condition inside and cell distribution in tumors (Figure 1E), and no significant tumor metastasis was observed in all groups. Further, necrotic characteristics including interstitial space enlargement (red arrows) exhibited more inside the tumor under combinational treatment.

### 2.2. Rapamycin Potentiates the Therapeutic Efficacy of Gefitinib in H1975 Cells by Inducing Autophagy

Given the augmented cell autophagy that emerged in cells under EGFR-TKI treatment, as many researchers demonstrated, we intended to kill gefitinib-resistant NSCLC cells via overwhelming autophagic turnover using the combination treatment of gefitinib and rapamycin. To confirm the role of cell autophagy in the synergistic effect of the combination treatment, we compared autophagy-related protein expression in H1975 cells under various conditions after 48 h of treatment, since autophagy is relatively late in the signaling transduction and starts peaking at 48 h after drug administration [24]. First, we analyzed lysates of drug-treated 2D and 3D cultures and tumor xenografts with LC3b and p62 (SQSTM1) antibodies, both of which have been identified as good indicators for autophagosome formation, and the content of LC3b-II is closely correlated with the number of autophagosomes [25]. In this study, it was observed that H1975 cells with rapamycin and the combination treatment expressed high levels of LC3b and p62 compared with gefitinib treatment (Figure 2A and Appendix A), indicating an accumulation of autophagosomes in cells, and the synergistic effect of gefitinib and rapamycin was closely associated with cell autophagy induction.

Surprisingly, gefitinib treatment on H1975 cells decreased cell autophagy levels compared to the control group (Figure 2A). These results indicate an exception of the cell autophagy response to EGFR-TKI treatment in H1975 cells. Therefore, we examined LC3b expression in 2D and 3D cultures. Western blot analysis was performed after exposure to gefitinib at a concentration of 0, 5 μMand 50 μM for 48 h. When we kept β-Actin expressed at a similar level, the relative expression of LC3b in 3D control presented significantly decreased expression levels in the 3D cultures at all dosages compared to 2D cultures (Figure 2B). Moreover, gefitinib treatment at an extremely high dose (50 μM) indeed increased cell autophagy induction compared to that at 5 μM, suggesting the role of cell autophagy in EGFR-TKI resistance is highly conditional. To investigate whether the augmented autophagy finally leads to cell death, we checked apoptosis-associated protein PARP and Caspase-3 expression in 2D cultures and tumor tissue. In general conditions autophagy crossregulate cell apoptosis. Appendix A showed that cleaved PARP and Caspase-3 increased significantly in cells under drug combination therapy, indicating elevated cell death.

As the target protein of gefitinib, EGFR plays a vital role in EGFR-TKI therapy. Thus, to further define other potential mechanisms responsible for the synergic inhibition effect, we explored whether obvious variations in EGFR protein expression were involved. The immunoblotting and immunofluorescence (Figure 2C, EGFR: upper group; LC3b: lower group) analysis was performed on cells and tumor xenografts. Our data indicated that an accommodated treatment of gefitinib (5 μM) decreased EGFR expression, whereas rapamycin or a combination of gefitinib and rapamycin treatment could reverse the situation (Figure 2A,C), which is consistent with the cell autophagy-associated protein changes.

### 2.3. Construction and Characterization of Aptamer-Functionalized Nanoparticles

To improve the delivery efficiency of combination drugs in advanced NSCLC cells, we designed a drug administration strategy based on nanomedicine. We co-encapsulated rapamycin and gefitinib into chitosan nanoparticles (NP) at a reasonable ratio to tune the cellular uptake; we then coated the NP with anti-EGFR aptamers to deliver drugs preferentially to the tumor site and to minimize non-targeted delivery. The anti-EGFR NP (NP-Apt) synthesis and its application in drug delivery are illustrated in Figure 3A. The chemical synthesis of NP and chemical surface modification of aptamers is based on ionic crosslinking and the EDC/NHS method. NP and NP-Apt displayed no significant diameter variation (Figure 3B). The zeta potential of NP-Apt decreased compared to NP because of the negatively charged DNA aptamer modification (Figure 3C). Transmission electron microscopy (TEM) revealed that the exact diameter of NP and NP-Apt are 80–100 nm (Appendix A and Figure 3D). The successful conjugation of aptamers with NP was verified through agarose gel analysis, as shown in Figure 3E. The mixture of aptamer and NP (first lane) and free aptamer (fourth lane) presented one band on the gel; NP alone did not show any bands (third lane), whereas NP-Apt remained at the loading site (second lane). Drug loading efficiency and stability of nanoparticles were defined by detecting unloaded drugs through UV–visible spectroscopy (Figure 3F,G). We evaluated gefitinib and rapamycin concentration at 343 and 283 nm, which is their maximum absorption wavelength, respectively. The linear regression curve suggested that the UV–visible absorption is positive correlated to the drug concentration (Appendix A). Additionally, our results showed that the gefitinib and rapamycin loading efficiencies were 30% and 80%, and the loaded rapamycin was stable in PBS at 37 °C for at least 48 h, whereas the loaded gefitinib was releasing gently and left 50% after 48 h.

### 2.4. Aptamer-Functionalized Nanoparticles Potentiate the Delivery Efficiency and Toxicity of Drug Combinations In Vitro

To visualize targeted drug delivery of NP-Apt, we constructed a co-cultivated cell system that contains H1975-mCherry cells (stably expressing mCherry fluorescent protein) and human embryonic kidney (HEK) 293T cells at a 1:1 ratio. The red fluorescence of mCherry made H1975 cells distinguishable under confocal microscope observation, and the profile of HEK 293T cells that hardly express receptors on the cell surface serves as a perfect control cell in this study. Meanwhile, we constructed fluorescein isothiocyanate (FITC)-tagged NP-Apt, and further spotted it bound mostly to target cells and rarely to non-target cells (Figure 4A, merge). Meanwhile, the CCK-8 assay demonstrated that targeted drug delivery (NP-Apt [gefitinib + rapamycin]) enhanced the cell toxicity of the combination therapy in H1975 cells (Figure 4B, left group), but caused no harm to non-target cells (Figure 4B, right group). The in vitro assay in the H1975 and 293T cell lines did not reveal any cytotoxicity of empty carriers (NP, NP-Apt) at the administrated concentration. We also observed activated cell autophagy activation in H1975 cells under nanomedicine treatment, although we reduced the dosage of each component. It is worth mentioning that our data showed that the anti-EGFR aptamer modified on NP is capable of activating cell autophagy and harmless to cell viability in an appropriate dosage (Figure 4C).

## 3. Discussion

Gefitinib’s reversible suppression of *EGFR* mutant tumors has gained global approval since its first in-human dosing in 1998, but acquired resistance ultimately occurs, and this has become a major restriction in advanced NSCLC [26]. Although novel EGFR-TKIs are being developed, their irreversible binding or broader inhibition of the HER family still has obstacles, such as acquired resistance and severe side effects [27,28]. In this regard, we aim to demonstrate an effective therapeutic way to overcome gefitinib resistance and eliminate advanced NSCLC cells effectively. This study was carried out utilizing advanced human NSCLC cell line H1975, which harbors both activating EGFR mutation (*L858R*) and secondary mutation (*T790M*) on the EGFR tyrosine kinesis domain. Some studies have shown that EGFR-TKI treatment can induce cell autophagy, and this is considered a self-protective process for cells from being damaged by EGFR-TKI treatment [12]. However, cell autophagy is a double-edged sword controlling cell fate in a precise way: it supports cell survival via digestion and recycling, but in certain physiological and pathological conditions, it leads to cell death, characterized by cytoplasmic vacuolation. Therefore, we speculated that a potent cell autophagy activator, rapamycin, could be a component of drug combination therapy to induce autophagic cell death in advanced NSCLC cells. In this study, we validated the high therapeutic potential of concomitant treatment with gefitinib and rapamycin on 2D/3D cultures and tumor xenografts (Figure 1A,B,D). Although the CI in 2D cultures suggested that the drugs are additive, we still believe rapamycin is an effective sensitizer of gefitinib in inhibiting the growth of H1975 cells in vitro and in vivo since 3D cultures are better in vitro models. Further, SynergyFinder 2.0 calculation (Appendix A) also suggested gefitinib and rapamycin are actually synergistic.

To define the role of cell autophagy in H1975 cells under the treatment of gefitinib, rapamycin, and the combination of gefitinib and rapamycin, we examined cell autophagy-related protein expression under different treatments. Increased LC3b-II in H1975 cells suggested that the combination of gefitinib and rapamycin induced cell autophagy significantly compared with gefitinib mono-treatment, whereas a slight decrease in LC3b-II also suggested that gefitinib caused downregulation of cell autophagy (Figure 2A). Meanwhile, p62 showed consistent upregulation with LC3b-II expression rather than being selectively degraded by autophagy, suggesting that autophagosomes are accumulated in cells instead of being fused with lysosomes and degraded after 48 h of treatment. The expression of cleaved PARP and Caspase-3 at 89 and 17 kDa (Appendix A) confirmed that the combination therapy successfully induced H1975 cells to death. Taken together, our data demonstrated that rapamycin mono-treatment and the combination of gefitinib and rapamycin both elevated cell autophagy levels, whereas rapamycin mono-treatment failed in cancer cell elimination. Thus, whether H1975 cell death could be classified as cell autophagic cell death remains unclear. We can only reasonably conclude that cell autophagy is closely related to the drug combination therapeutic effect, other cell activities’ inter-connection, or an alteration of cell tolerance of autophagy under gefitinib treatment could be reasonable.

Additionally, there was a huge cell autophagy variation between 2D cultures and 3D cultures. A complete inhibition of autophagy was observed in cells grown in the 3D culture system compared with cells grown in the 2D culture system (Figure 2B). Our data suggested that the gefitinib IC_50_ values in 3D cultures (10.64 μM) were slightly higher than in 2D cultures (6.95 μM); in addition, a high concentration of gefitinib (50 μM) inhibited 2D cell proliferation enormously and 3D cell proliferation poorly (Figure 1A,B). These observations also mirror the current cancer studies in that 3D cultures often display lower cell proliferation and higher resistance to chemotherapeutic drugs [29]. Since 3D cell culture systems are widely considered an approach closer to complex in vivo conditions, we concluded that cell autophagy was actually inhibited in H1975 cells and that gefitinib treatment also failed to induce cell autophagy induction.

Our mechanistic studies have suggested that H1975 expresses low levels of EGFR when treated with gefitinib, and that there is a clear positive correlation between EGFR and cell autophagy induction (Figure 2A,C). Recently, Song et al. reported a novel LC3b–EGF–EGFR regulation in NANOG^+^ tumor cells and demonstrated that LC3b contributed to the secretion of EGF, thereby activating EGFR signaling [30]. Accordingly, we suggest that increased LC3b may lead to increased EGF–EGFR signaling and then relief gefitinib resistance in H1975 cells (Figure 5). To be specific, less EGFR expression made gefitinib useless, the arisen of other receptors on the cell membrane such as MET also contributes to gefitinib resistance in NSCLC cells [7]. When H1975 cells were treated with rapamycin and gefitinib simultaneously, rapamycin could first inhibit mTOR, lead to LC3b overexpression and thereby enhance EGF secretion. As the receptor of EGF, more EGFR will exist on the cell membrane, and the EGFR–AKT signaling pathway will become significant for cell survival. Next, gefitinib will inhibit EGFR and prevent cell proliferation. In the end, cells will die of autophagy-related mechanisms.

Although the mechanism remains unclear, the proposed gefitinib and rapamycin drug combination therapy has been validated to be efficient in gefitinib-resistant tumor regression. For a better clinical application, we investigated NP-Apt as a desirable delivery system. Its properties of co-encapsulation and target drug delivery strategy could tune the dose of drugs and coordinate drug biodistribution. Chitosan is cationic and mucoadhesive, the ligand-receptor interaction could also mediate endocytosis. All the aforementioned characteristics of NP-Apt-enabled drugs to remain at the target site for a longer period.

TEM images of NP-Apt revealed its spherical morphology with a diameter less than 100 nm (Figure 3D), which is beneficial to drug delivery in terms of enhanced permeability, retention, and cellular uptake through electronic attraction with the cell membrane. Meanwhile, the good stability of NP-Apt-enabled drugs (gefitinib + rapamycin; Figure 3G) to be delivered to the tumor area without significant leakage. The target delivery was verified through confocal observation of FITC-tagged NP-Apt in target cells and control cells. Co-localization of red, green, and blue fluorescence indicated NP-Apt could be accumulated around target cells (Figure 4A). following cell viability assay of NP-Apt (gefitinib + rapamycin) also confirmed its therapeutic efficiency in target and non-target cells (Figure 4B). The Western blotting analysis proved the mechanism of modified nanomedicine could activate cell autophagy as drug combinational treatment did, indicating a potential cellular function of anti-EGFR aptamer in cell autophagy activities (Figure 4C), which is worth further investigation. Taken together, the results indicated that NP-Apt (gefitinib + rapamycin) is promising for the improvement of the drug combination therapy in vivo.

In conclusion, we described a nanomedicine that can overcome EGFR-TKI resistance in NSCLC cells based on a drug combination strategy. Gefitinib and rapamycin synergistically inhibit advanced human NSCLC cell line (H1975, harboring *EGFR T790M* secondary mutation) proliferation and tumor formation by upregulating cell autophagy activities, thereby inducing cell autophagy–EGFR positive feedback regulation. Meanwhile, anti-EGFR aptamer-decorated chitosan nanocarriers co-encapsulated gefitinib and rapamycin exalted multi-drug administration accuracy. Our data suggest that rapamycin could be a preferred sensitizer of gefitinib in acquired gefitinib resistance, whereas the crosstalk between cell autophagy and EGFR signal transduction and the potential cellular function of the aforementioned anti-EGFR aptamer is still worth studying.

## 4. Materials and Methods

### 4.1. Drugs and Chemicals

Cell counting kit 8, gefitinib hydrochloride, and rapamycin were obtained from Med Chem Express (Beijing, China). A 3D cell viability assay was purchased from Progema (Beijing, China). Antibodies against p62, LC3b, GAPDH, and horseradish peroxidase (HRP)-conjugated goat anti-rabbit IgG were purchased from Cell Signaling Technology; those against EGFR, and β-Actin, PARP, Caspase-3, cleaved Caspase-3 were obtained from Abclonal (Wuhan, China). Matrigel was purchased from Corning. Chitosan (deacetylation degree ≥ 95%, biotechnology level) and sodium tripolyphosphate (TPP) were purchased from Macklin (Shanghai, China).

### 4.2. Cell Culture

Two-dimensional (2D)- and three-dimensional (3D)-cultured H1975 cells were maintained in a cell culture incubator at 37 °C with 5% CO_2_ with RPMI 1640 medium (Thermofisher, Shanghai, China) supplemented with 10% fetal bovine serum (FBS, Tsingmu Biotechnology, Wuhan, China). Three-dimensional cultures were formed through mixing H1975 cells with Matrigel (Corning, NY, USA) using microfluidic technology, as we previously reported [21].

### 4.3. Cell Viability Assay

Two-dimensional and 3D cell viabilities were assessed using a CCK-8 and 3D cell viability assay (Promega, Madison, WI, USA), respectively. Two-dimensional cultures (8000 cells per well) and 3D cultures (one cell spheroid per well) were seeded in the 96-well plate and treated with various concentrations of gefitinib, rapamycin, gefitinib + rapamycin, NP, NP-Apt, NP (gefitinib + rapamycin), and NP-Apt (gefitinib + rapamycin) for 48 h. Next, CCK-8 solution and 3D cell viability assay reagent were added to each well, respectively. For 2D cultures, absorbance was measured at 450 nm (Thermo, Waltham, MA, USA) after 1 h of incubation at 37 °C. For 3D cultures, the luminescence signal (Thermo, USA) was read after 30 min of incubation at room temperature.

### 4.4. Clonogenic Assay

Two-dimensional-cultured H1975 cells were seeded into six-well plates at a density of 2000 cells per well, and they were cultured in the medium with 5% FBS at 37 °C in a humidified atmosphere containing 5% CO_2_. The cells were treated with medium, gefitinib (5 μM), rapamycin (10 μM), and gefitinib (5 μM) + rapamycin (10 μM). After 15 days, cells were washed twice with PBS and then fixed with paraformaldehyde for 15 min. The cells were stained with 0.1% crystal violet for 30 min and then visualized under a camera (Nikon, Tokyo, Japan).

### 4.5. In Vivo Tumorigenesis Assay

The H1975 mouse lung cancer model was established for the following in vivo evaluation. Female BALB/C nude mice (4–6 weeks of age) were purchased from Guangdong Medical Laboratory Center, and all mice were handled under the protocols approved by the Institutional Animal Care Committee. In this study, 5 × 10^6^ H1975 cells were suspended in 100 μL PBS for subcutaneous injection. After 7 days of tumor formation, the mice were randomized into four groups (control, gefitinib, rapamycin, and gefitinib + rapamycin). Gefitinib (150 mg/kg orally by gavage) and rapamycin (2 mg/kg intraperitoneal injection) were administered every 2 days and continued for 15 days. Body weight and tumor size were recorded, tumor volume was calculated as 0.5 × length × width^2^, and the animal experiment was repeated three times, with eight mice for each group in total. Body weight and tumor size were recorded every 2 days until the xenograft tumor was stripped for further study, and no significant body weight decrease was observed.

### 4.6. Histopathology Study

The tissues were fixed with 4% paraformaldehyde and sliced into 8 μm sections after being embedded with paraffin. The hematoxylin and eosin (H&E) staining followed the standard procedure and was observed using optical microscopy (Nikon, Tokyo, Japan).

### 4.7. Western Blotting

H1975 with different treatments were cultured in the 2D/3D cell culture system for 48 h and then lysed with RIPA lysis (Beyotime, Haimen, China) buffer at 4 °C. The DNA or Matrigel was removed via centrifugation (ThermoFisher, Waltham, MA, USA) at 10,000 rpm for 15 min. Total protein was quantified with a BCA Protein Assay Kit (ThermoFisher, USA) and then mixed with loading buffer and boiled at 100 °C for 10 min. Protein extracts (30 μg) were resolved and separated by 12.5% sodium dodecyl sulfate-polyacrylamide gel electrophoresis (SDS-PAGE) and then transferred to nitrocellulose membranes. The membrane was incubated in the sequence of blocking solution (5% non-fat milk) for 1 h, primary antibody EGFR, β-Actin, p62, and LC3b solution for 3 h, and HRP-tagged secondary antibody solution for 1 h at room temperature with gentle rocking. The nitrocellulose membranes were washed three times with phosphate-buffered saline with 0.05% Tween-20 (PBST) buffer before every incubation. Finally, the membrane was incubated with enhanced chemiluminescence for 5 min, and the blot was imaged using an X-ray imaging system (Bio-Rad, Hercules, CA, USA). Every experiment was performed at least 3 times. The relative ratio of EGFR, p62, and LC3b-II to β-Actin was evaluated using Image J software.

### 4.8. Immunofluorescence

Sliced tumor tissue samples were fixed in 4% paraformaldehyde for 30 min. After being blocked with 3% BSA/PBS, the cells were stained with primary antibodies (EGFR, LC3b), followed by staining with the corresponding fluorophore-conjugated secondary antibodies. The cell nucleus was stained with DAPI for 10 min. Samples were mounted on coverslips and examined under a confocal microscope (Olympus, Tokyo, Japan).

### 4.9. Aptamer-Functionalized Drug-Loaded Chitosan Nanoparticle Preparation

Chitosan nanoparticles (NP) were prepared by ionic crosslinking with sodium tripolyphosphate (TPP) according to the methodology of Alessandro et al. [31]. Specifically, gefitinib and rapamycin were dissolved into 10 mL of chitosan solution (2 mg/mL dissolved in 1% acetum, pH 5), and 3 mL of TPP (2 mg/mL) was added to the system in a dropwise fashion under room temperature with magnetic stirring at 800 rpm for 30 min.

Aptamer-modified chitosan nanoparticle (NP-Apt) was synthesized using the EDC/NHS conjugation method. In short, 1 nmol of carboxyl-modified anti-EGFR aptamer [20] (5′-TAC CAG TGC GAT GCT CAG TGC CGT TTC TTC TCT TTC GCT TTT TTT GCT TTT GAG CAT GCT GAC GCA TTC GGT TGA C-3′) was conjugated to 8 mg of NP by adding EDC and NHS and stirring at room temperature for 3 h. The final concentrations of EDC and NHS were 50 and 5 mM, respectively. The final product—namely NP, NP (gefitinib + rapamycin), NP-Apt, or NP-Apt (gefitinib + rapamycin)—was obtained via centrifuging (ThermoFisher, USA) at 10,000 rpm at 4 °C for 10 min, washing twice with distilled water, and suspending the product suspended in cell culture medium for future study.

### 4.10. Characterization of NP and NP-Apt

Particle size and electrostatic potential were measured using a nano dynamic light scattering analyzer (Malvern, Worcestershire, UK). The shape and surface morphology of NP-Apt were observed using transmission electron microscopy (TEM, FEI Tecnai G2 Spirit, Hillsboro, OR, USA).

### 4.11. Agarose Analysis

Agarose gel electrophoresis was used to confirm the conjugation of aptamer binding on the NP surface. Ten microliters of the mixture of Apt and NP, NP-Apt, NP, and free aptamer was mixed with loading buffer and then loaded on the agarose gel stained with SYBR green. The electrophoresis picture was obtained using the Bio Rad ChemiDoc MP imaging system (Hercules, CA, USA).

### 4.12. Drug Loading Efficiency and Stability

Drug concentration was detected using a UV–visible spectrophotometer (Agilent, Santa Clara, CA, USA) at 343 and 283 nm after NP-Apt (gefitinib + rapamycin) was removed from the system through centrifuging. Drug loading efficiency was calculated by subtracting the unencapsulated drug amount from the original amount. We detected drug-loading stability by monitoring an in vivo drug release: NP-Apt (gefitinib + rapamycin) was suspended in PBS at 37 °C, and the concentration of encapsulated drug was calculated by the detection of drug released concentration at specific times (6, 12, 24, 48 h).

### 4.13. Cellular Uptake and Target Delivery of NP-Apt

Cells from the 293T and H1975 lines labeled with a red fluorescent protein (H1975-mCherry) were cocultured and incubated with FITC-tagged NP-Apt (NP-Apt-FITC) for 3 h. Then, the cell nuclei were counterstained with DAPI. The specimens were examined under a confocal laser scanning microscope (Olympus, Japan).

### 4.14. Statistical Analysis

All data were repeated at least three times and presented as the mean ± standard deviation (SD). The comparison was performed using a two-tailed paired Student’s *t*-test. (* *p* < 0.05, ** *p* < 0.01, and *** *p* <0.001). The combinatorial index (CI) was calculated as CI = (Ea + Eb − Ea * Eb)/Eab, where E means cell viability of compound a or b or the ab (combination of both). The combination of drugs with a 0.9 < CI < 1.1 was considered additive, whereas the combination of drugs with a CI < 0.9 was considered synergistic. SynergyFinder was used to calculate dose responses and the HSA synergy score. We applied the HSA model to quantify the degree of synergy, and the scores were determined as the average excess response due to drug interactions [23].

## Figures and Tables

**Figure 1 ijms-23-08025-f001:**
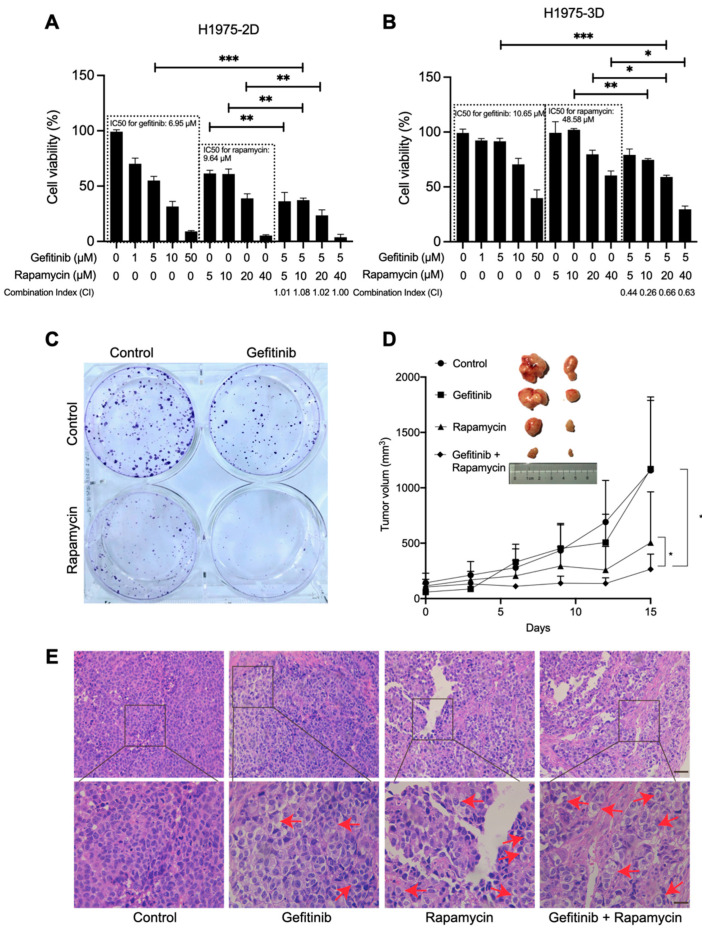
Combination therapy of rapamycin and gefitinib attenuates H1975 cell growth. (**A**) CCK-8 assay of 2D-cultured H1975 cells under the treatment with gefitinib, rapamycin, and their combination for 48 h at a gradient concentration, as indicated. ** *p* < 0.01 and *** *p* < 0.001. (**B**) Cell viability assay of 3D cultures under the treatment with gefitinib, rapamycin, and their combination for 48 h at a gradient concentration, as indicated. ** *p* < 0.01 and * *p* < 0.05. (**C**) Clonogenic assay performed in 2D-cultured H1975 cells treated with gefitinib (5 μM) and/or rapamycin (10 μM). (**D**) Photograph of harvested H1975 tumor xenografts after 15 days of treatment and tumor xenograft growth within 15 days of treatment. (Control: phosphate-buffered saline (PBS); gefitinib: 150 mg/kg; rapamycin: 2 mg/kg). * *p* < 0.05. (**E**) Light microscopic analysis of H&E-stained H1975 tumor xenograft sections after 15 days of treatment with different therapies. Red arrows indicate cell necrosis.

**Figure 2 ijms-23-08025-f002:**
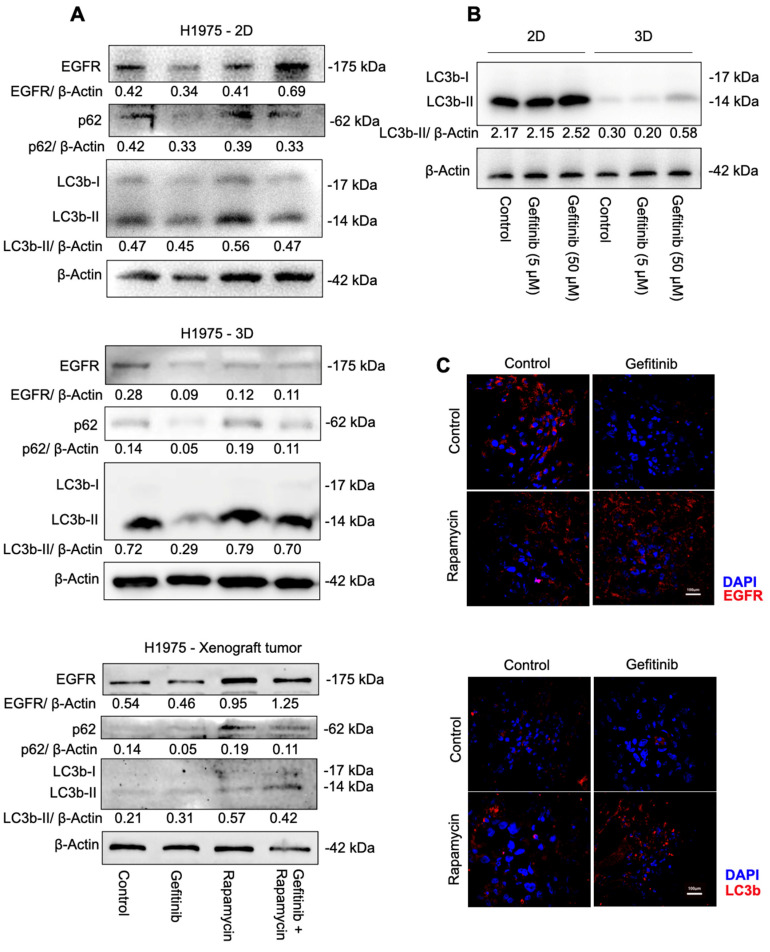
Autophagy activity in H1975 cells and xenografts under treatment with gefitinib, rapamycin, and their combination. (**A**) Immunoblotting and quantities analysis of p62, LC3b, and β−Actin protein expression in 2D cultures (gefitinib: 5 μM; rapamycin: 10 μM), 3D cultures (gefitinib: 5 μM; rapamycin: 20 μM), and xenograft tumors (gefitinib: 150 mg/kg; rapamycin: 2 mg/kg). (**B**) Immunoblotting analysis of LC3b and β−Actin protein expression in 2D cultures, 3D cultures. (**C**) Immunofluorescence analysis of EGFR (upper) and LC3b (lower) protein expression in H1975 tumor xenografts treated with gefitinib: (150 mg/kg) and/or rapamycin (2 mg/kg).

**Figure 3 ijms-23-08025-f003:**
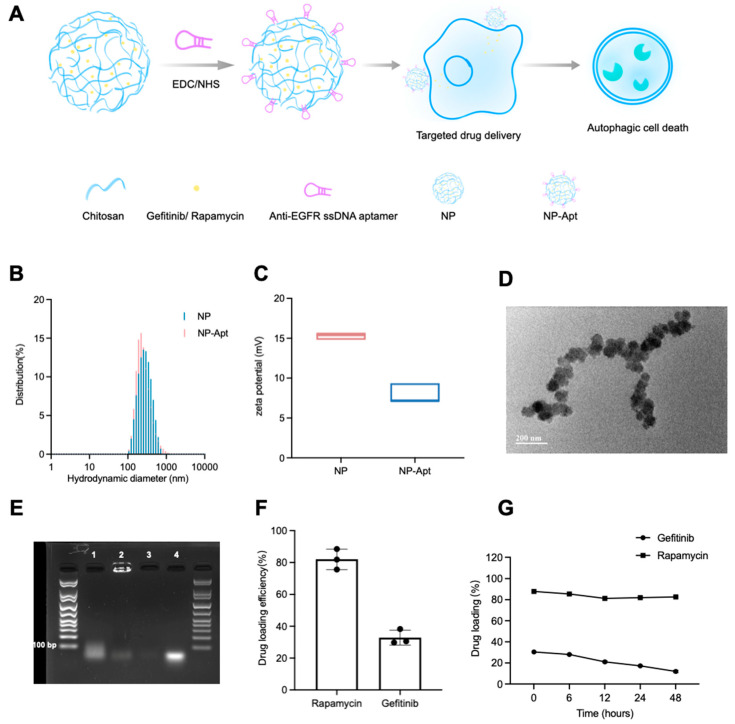
Construction and characterization of NP and NP-Apt. (**A**) Schematic illustration of gefitinib and rapamycin-loaded NP-Apt (NP-Apt [gefitinib + rapamycin]). (**B**) The hydrodynamic size distribution of NP and NP-Apt. (**C**) Zeta potential analysis of NP and NP-Apt. (**D**) TEM picture of NP-Apt. (**E**) Agarose gel electrophoresis. 1. The mixture of Apt and NP; 2. NP-Apt; 3. NP; 4. Free aptamer. (**F**) Drug loading efficiency of gefitinib and rapamycin. The maximum absorbance of gefitinib and rapamycin was 343 and 283 nm, respectively. (**G**) Stability of NP-Apt (gefitinib + rapamycin) at 37 °C in PBS.

**Figure 4 ijms-23-08025-f004:**
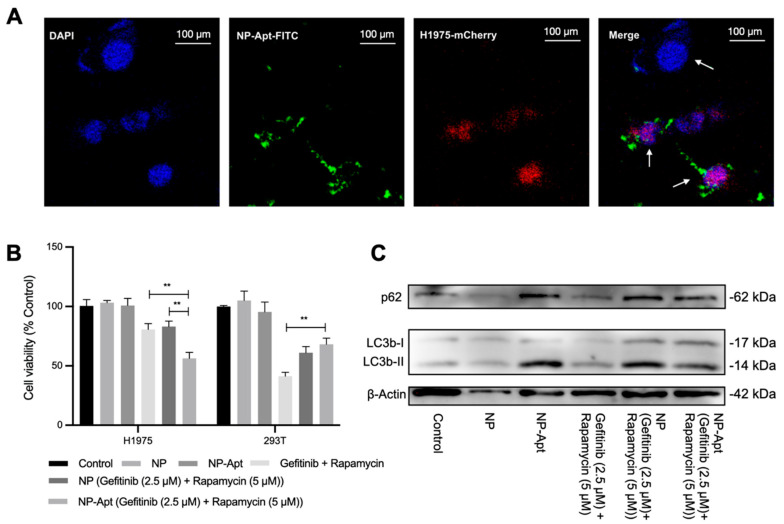
Target delivery and toxicity of NP−Apt (gefitinib + rapamycin). (**A**) Selective cell uptake of NP−Apt. H1975 cells 293T cells were co−cultured in one dish. H1975 cells displayed blue and red fluorescence due to the DAPI staining and mCherry transfection. The 293T cells displayed only blue fluorescence. NP−Apt were conjugated with FITC showing green fluorescence. White arrows indicate 293T and H1975 cells. (**B**) Cell viability assay 48 h of H1975 cells and 293T cells after various treatments: control (DMEM), NP, NP−Apt, gefitinib + rapamycin, NP (gefitinib + rapamycin), NP−Apt (gefitinib + rapamycin). The dosages of gefitinib and rapamycin were 2.5 and 5 μM. All data are representative of at least three independent experiments. ** *p* < 0.01. (**C**) Immunoblotting analysis of p62 and LC3b protein expression in H1975 cells under different treatments.

**Figure 5 ijms-23-08025-f005:**
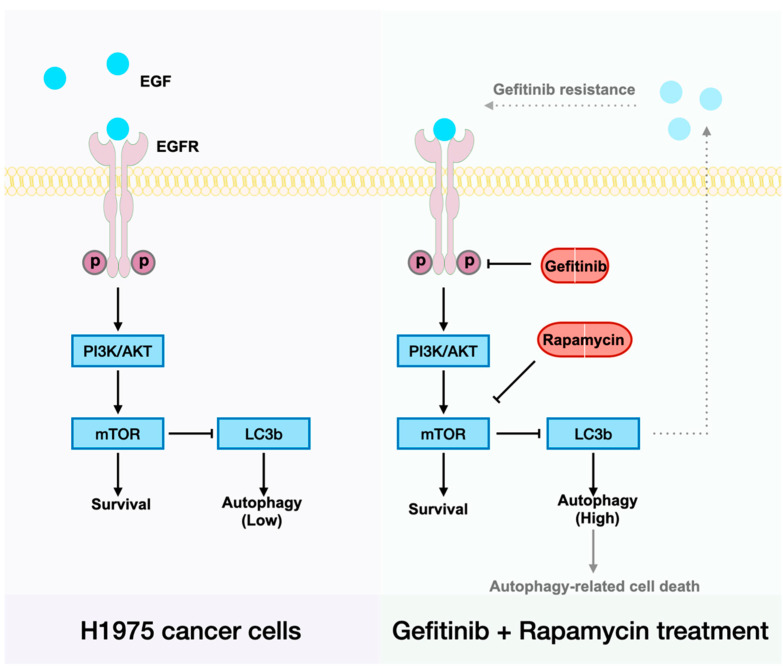
Schematic illustration of the cellular mechanism of drug combination therapy in gefitinib-resistant cells. Dotted lines and translucent words/figures denote points that have not yet been validated.

## Data Availability

Not applicable.

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
