# Peer review of "Targeted Co-Delivery of Gefitinib and Rapamycin by Aptamer-Modified Nanoparticles Overcomes EGFR-TKI Resistance in NSCLC via Promoting Autophagy"

_ijms, 2022, doi:10.3390/ijms23148025_

Round 1
Reviewer 1 Report
The manuscript of Yuhong Liu and colleagues presents results on combination of EGFR-TKI and autophagy inducer rapamycin in attempt to overcome resistance of NSCLC cells to TKI treatment. The results are of certain novelty and scientific interest. In my opinion, they can be published after minor revision.
- The main issue is that the mechanism of the positive feedback between autophagy and EGFR remains unclear, especially in the part about hyperactivation of EGFR-AKT signaling in combination treated cells and its relation with cell death. I recommend authors to revise the explanation of their hypothesis and make its mire reader-friendly.
- The discrepancy in results of synergism assessment by different methods should be commented: in 2D culture the synergistic effect was rejected in Fig. 1A but supported in Fig. S2AB.
- The authors state an increase in p62 and LC3b-II (Lines154-156) in rapamycin and combination treated cells compared with gefitinib; however, this is not true for combination treated 2D culture. Please, be accurate in the conclusion.
- Similar situation is with NP-Apt stability (Lines 209-210). The gefitinib loading in NP-Aps decrases by more than two times in 48 hours. The fact shouldn’t be just ignored.
- Please, correct the legend for Fig. 2 (2B), it is not corresponds to the data presented. Also, the pronounced discrepancy in the relative amount of LC3b-II in the untreated control 3D culture in Fig. 2A (0.72) and 2B (0.3) should be explained.
- The authors state decreased EGFR expression under gefitinib treatment (immunofluorescence in Fig.2C). But, the images present decreased EGFR under rapamycin. The same situation is with LC3b. Is the labeling correct?
- Wavelengths chosen for measurement of concentrations of gefitinib and rapamycin must be explained.
- The information about names of most of the equipment as well as Abs used in the study is omitted in the present version of the article.
Reviewer 2 Report
The authors have written a very interesting paper on the synergistic effects of rapamycin with gefitinib. While the results are interesting there is some work to be done on the presentation. Attached is a document of comments to be looked at. Many are minor corrections. The most major comment and work that needs to be done arises from the fact that, while the English is fairly well written in the sense of grammar etc, often the explanation of the results and their significance are highly lacking making it difficult for the reader to understand without heavily analysing the figures by themselves. This must be improved to make the results and conclusions more accessable to the readers.
· 94-102 seem more like intro where you explain the work not results.
· Why did the authors choose 48 hours. Is this standard in the literature?
· Line 107 what was the 2d IC50?
· Figure 1. why did the authors choose hold constant gefitinib at 5 while changing rapamycin? The CIindex in Fig 1 A was highest at 5 and Please explain this choice more clearly the quote below
· Figure 1 c…. also here. Why were 5 and 10 chosen. This reviewer does not see a full logic in the selection of the concentrations. Please explain as to why the values optimized in supplemental were not used for this 2d method.
“For 2D/3D cultures, the IC50 value for the combination therapy is supposed to be 118
gefitinib (5 μM) + rapamycin (1 μM) and gefitinib (10 μM) + rapamycin (20 μM), respec- 119
tively (Figure S2A, C).”
· Fig 1e indicate what the arrows are showing.
· These sentences seem to be incomplete:
H&E staining displayed the 125
condition inside and cell distribution tumors (Figure 1E), and no significant tumor metas- 126
tasis was observed. Besides, necrotic characteristics including interstitial space enlarge- 127
ment exhibited inside the tumor under combinational treatment.
Line 147 “detected” seems to be the wrong word here.
· This also goes in the introduction and not the results.
tophagosomes, severing as autophagosomal surface protein and an autophagic substrate 150
protein, respectively. Their interaction mediates the inclusion formation during the au- 151
tophagy process, which eventually leads to a cell's demise [24]. LC3b presents two bands 152
(LC3b-I and LC3b-II) in western blot analysis, and the content of LC3b-II is closely corre- 153
lated with the number of autophagosomes [25].
· Fig 2 needs a bar graph of the quantification of these bands (with programs like image J) to show a percent or fold decrease to better understand the results.
Authors report the %EE of the two drugs. But how does this equate to the amount of each? Does it come out to a ratio similar to those used in previous experiments????

Round 2
Reviewer 2 Report
the authors have adressed most of the major points to clarify the paper.